# Mechanochemical synthesis of organoselenium compounds

Shanshan Chen[1], Chunying Fan[1], Zijian Xu [2], Mengyao Pei[1], Jiemin Wang[1], Jiye Zhang[1], Yilei Zhang [3], Jiyu Li[4], Junliang Lu[4], Cheng Peng [1] ✉ & Xiaofeng Wei [1] ✉

We disclose herein a strategy for the rapid synthesis of versatile organoselenium compounds under mild conditions. In this work, magnesium-based selenium nucleophiles are formed in situ from easily available organic halides, magnesium metal, and elemental selenium via mechanical stimulation. This process occurs under liquid-assisted grinding (LAG) conditions, requires no complicated pre-activation procedures, and operates broadly across a diverse range of aryl, heteroaryl, and alkyl substrates. In this work, symmetrical diselenides are efficiently obtained after work-up in the air, while one-pot nucleophilic addition reactions with various electrophiles allow the comprehensive synthesis of unsymmetrical monoselenides with high functional group tolerance. Notably, the method is applied to regioselective selenylation reactions of diiodoarenes and polyaromatic aryl halides that are difficult to operate via solution approaches. Besides selenium, elemental sulfur and tellurium are also competent in this process, which showcases the potential of the methodology for the facile synthesis of organochalcogen compounds.

As an essential trace element for human beings, selenium manifests a wide range of physiological processes through in-corporation into over 25 selenoproteins as selenocysteine[1]. The selenium-containing compounds are also prevalent in pharmaceutical libraries[2–6], catalyst scaffolds[7–18], and material building blocks[19,20] (Fig. 1a). These important applications have triggered the development of robust, concise, and environmentally friend methods for their synthesis to access new chemical spaces. Whilst many synthetic methods including metal-catalyzed (Ni, Cu, Pd, Fe, Co, In, etc.) cross-coupling reactions, metal-free oxidative coupling reactions, as well as photochemical and electrochemical reactions have been developed for the synthesis of organoselenium compounds using preformed activated selenium reagents[21–36], direct utilization of elemental selenium in the synthesis is a more attractive strategy due to elemental selenium's low price, commercial availability, storage stability, odourlessness, and ease of handing. Nevertheless, the inert chemical property, low solubility in

organic solvent, and tendency to form transition-metal selenium clusters make direct use of selenium powder for efficient synthesis challenging (Fig. 1b)[37]. To date, the majority of these methods require large excess of polar solvents, long reaction time, and harsh conditions. In contrast with traditional solvent-based strategies, mechanochemistry allows efficient energy dispersion and mass transportation under solid-state condition and therefore provides an alternative strategy for green and sustainable synthesis[38–47]. Recently, the strategy has been applied to facilitate the oxidative addition of zero valent metals (such as Mg, Mn, Zn, Ca, etc) to organic halides, generating organometallic species for diverse nucleophilic transformations[48–60]. For instance, Ito, Kubota[60], and Bolm[61] groups independently demonstrated elegant examples of producing versatile Grignard reagents using mechanochemical strategy. Remarkably, in the work of Ito and Kubota[60], no degradation of air-sensitive Grignard reagents was observed even when the milling jar was briefly

[1]School of Pharmacy, Xi'an Jiaotong University, No. 76, Yanta West Road, Xi'an 710061, China. [2]Shanghai Synchrotron Radiation Facility, Shanghai Advanced Research Institute, Chinese Academy of Sciences, Shanghai 201204, China. [3]Department of Biochemistry and Molecular Biology, School of Basic Medical Sciences, Xi'an Jiaotong University Health Science Center, Xi'an, Yanta, China. [4]Xi'an Aisiyi Health Industry Co., Ltd, Xi'an 710075, China. ✉e-mail: pcheng@xjtu.edu.cn; xiaofeng.wei@xjtu.edu.cn

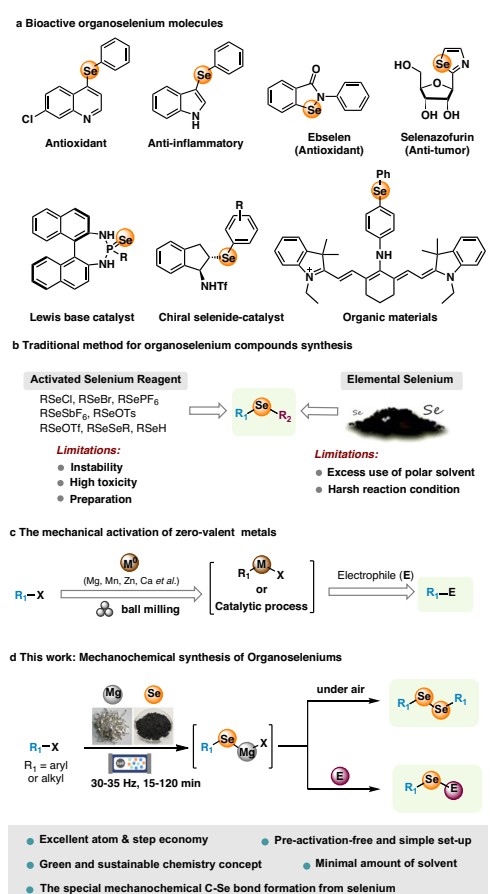

**Fig. 1 | Synthesis of organoselenium compounds. a** Functional organoselenium molecules; **b** Traditional method for organoselenium compounds synthesis; **c** The mechanical activation of zero-valent metals; **d** Mechanochemical route.

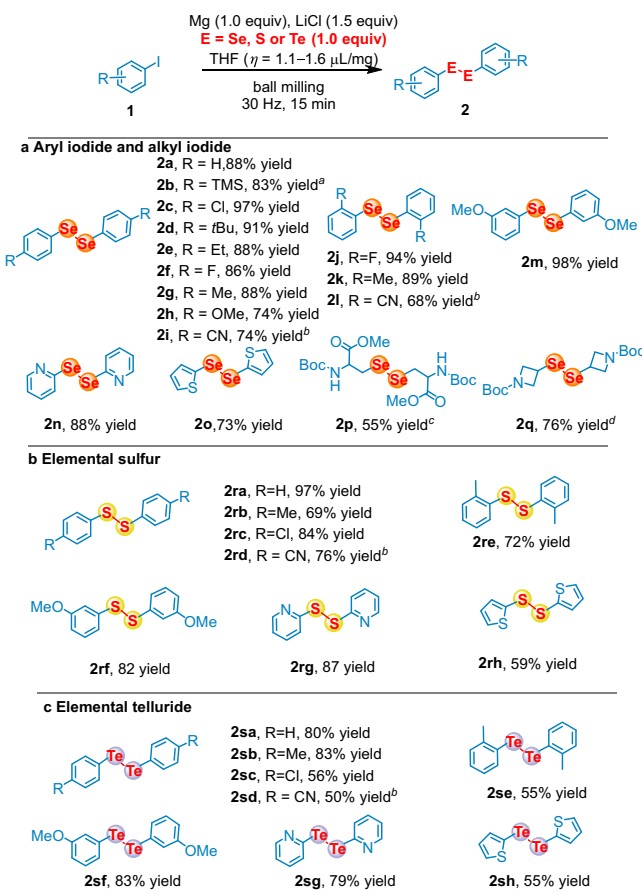

**Fig. 2 | Substrate scope for the synthesis of symmetrical dichalcogenides. a** Aryl iodide and alkyl iodide; **b** Elemental sulfur; **c** Elemental telluride. Reaction conditions: a stainless-steel milling jar (1.5 mL) and a stainless-steel ball (6 mm) were used; isolated yields; $\eta$ = V (liquid; μL) / m (reagents; mg). For details, see the Supplementary Information Condition A; [a]**1b** = 4-TMS-PhBr; [b]60 min, 35 Hz, Mg (1.5 equiv); [c]120 min, 35 Hz, Mg (0.5 equiv). For details, see the Supplementary Information Condition C; [d]30 min.

exposed to the air before adding an electrophile, highlighting the method's robustness.

In this work, we present a mechanochemical method for synthesizing organoselenium compounds which involves the in situ generation of magnesium-based selenium species through the straightforward process of mixing and grinding organic halides, magnesium, and elemental selenium. Notably, these species exhibit extreme sensitivity to both oxygen and water, leading to their complete conversion into symmetrical diselenides during the work-up procedure. Additionally, employing a one-pot process for the addition of electrophiles enables efficient synthesis of unsymmetrical monoselenides, which proceeds smoothly even in the presence of air. (Fig. 1d). We also achieve the successful preparation of magnesium-based organoselenium reagents from polyaromatic aryl halides and diiodoarenes. Notably, it's important to highlight that converting such substrates into organochalcogenides poses challenges when employing conventional solution-based methods. Near edge X-ray absorption fine structure (NEXAFS) spectroscopy is employed to analyze the generation of the magnesium-based organoselenium nucleophiles under mechanochemical conditions. The method can be extended to the straightforward synthesis of organic sulfur and tellurium compounds, suggesting its potential to serve as a highly practically foundation for the comprehensive synthesis of organochalcogen compounds.

## Results

As a proof of concept, we chose iodobenzene (**1a**) as a model substrate to be added to a 1.5 mL stainless-steel milling jar loaded with commercially available magnesium turnings (1.0 equiv relative to **1a**),

selenium powder (1.0 equiv relative to **1a**), LiCl (1.5 equiv relative to **1a**), THF ($\eta$ = 1.4 μL/mg), and one stainless-steel balls (diameter: 6 mm) in argon glovebox (For detailed screening of reaction parameters, see the Supplementary Information, Supplementary Table 2). The reaction was conducted using Retch MM400, while Retch MM500 was used in the case of higher vibration frequency (35 Hz). The reaction was monitored and the starting material was completely consumed within 15 min. To our surprise, instead of getting benzeneselenol, which was the predominant product in solvent-based activation of elemental selenium using Grignard reagents[62–64], diphenyl diselenide was obtained in 88% yield (Fig. 2a). Analysis of crude NMR showed only approximately 5% benzenselenol was generated, while potential side products such as biphenyl and diphenylselenide were not observed. The high chemoselectivity for diselenide showed the advantage of our strategy. Up to date, preparation of diselenide from elemental selenium generally requires a transition metal catalyst, harsh reaction conditions, and long reaction time[65–68], indicating our approach provides a solution for their facile synthesis.

Having established the optimal conditions, we next examined the scope of the reaction. Products were obtained in high yield for aryl iodides possessing electron-donating groups, halogen, and sterically hindered substituents at all *para*-, *ortho*-, and *meta*- positions (**2a, 2c-2h, 2j-2k, 2 m**, 74-98%). Aryl bromide (**1b**) also performed well under the optimized condition (**2b**, 83%). Heteroaryl iodides (**1n** and **1o**) were also competent under the standard reaction as the corresponding

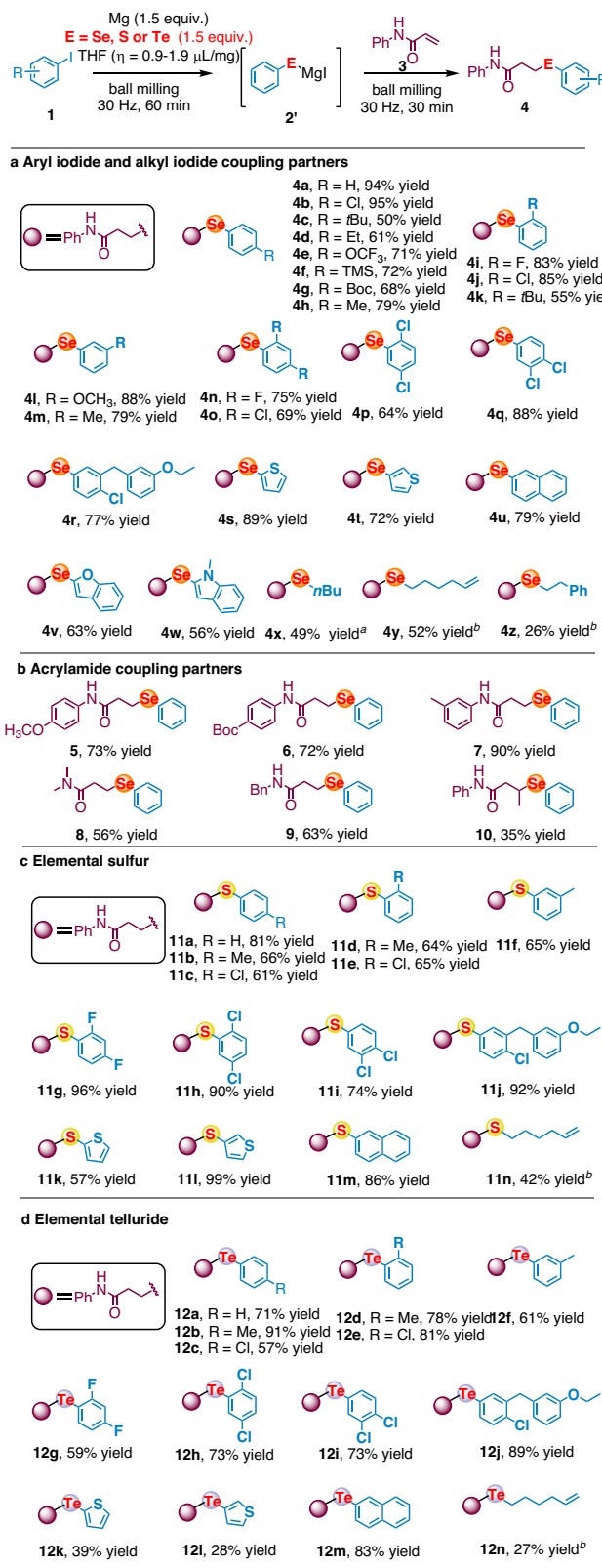

**Fig. 3 | Substrate scope for the synthesis of unsymmetrical monochalcogenides using acrylamides as acceptor. a** Aryl iodide and alkyl iodide coupling partners; **b** Acrylamide coupling partners; **c** Elemental sulfur; **d** Elemental telluride. Reaction conditions: a stainless-steel milling jar (1.5 mL) and a stainless-steel ball (diameter: 6 mm) were used; isolated yields; $\eta$ = V (liquid; μL) / m (reagents; mg). For details, see the Supplementary Information Condition D. [a]**3a** (0.5 mmol), Mg (5.0 equiv), Se (2.0 equiv), **1w** (2.0 equiv), THF ($\eta$ = 1.3 μL/mg) in stainless-steel milling jar (5.0 mL) with a stainless-steel ball (diameter: 10 mm); [b]**1y** = 6-bromo-1-hexene. Additional Mg (1.0 equiv) was added in the second step; for details, see the Supplementary Information Condition F.

in moderate to good yields (**2ra-2rh** and **2sa-2sh**). Due to the plastic feature, S$_8$ was generally considered inert against mechanical impact, while most of the kinetic energy deriving from the collisions between the balls and sulfur is consumed its deforming rather than for promoting its reactivity[69]. Although investigation in chalcogenation process via ball-milling strategy is raising considerable concern[70-73], direct construction of carbon-chalcogen bond from chalcogen element is rare. Herein, we report the mechanochemical synthesis of organochalcogenides using elemental chalcogen, which showed significant improvement of conversion compared with solution-based condition (Supplementary Information, Supplementary Table 2 and Table 3). Moreover, with modified procedure (Supplementary Information, Condition C), racemic amino acid derivative (**1p**) containing protonic functionality was also applicable, which showcased the compatibility of our method with protonic functionality, which is challenge in the process for preparing Grignard reagents. Besides, this method was applicable for heterocyclic alkyl iodide (**1q**), providing symmetrical alkyl diselenide **2q** in good yield.

Further, one-pot mechanochemical process was designed to convert elemental selenium directly to unsymmetrical monoselenides (Fig. 3a). We assumed the existence of reaction equilibrium between symmetrical diselenide (**2**) and magnesium-based selenium nucleophiles (**2'**) in the presence of excess amount of magnesium under ball-milling condition, therefore the symmetrical diselenide (**2**) would gradually transform to monoselenides in the presence of electrophile. As expected, when N-phenylacrylamide (**3a**) was added in air, the subsequent ball-milling reaction afforded conjugate addition product **4a** in 94% yield (For detailed screening, see the Supplementary Information, Supplementary Table 3). Our protocol was found to be general and robust with remarkable functional group tolerance even in the presence of protonic *NH* functional group of the acryl amides substrates. A broad range of electronic and sterically differentiated substituted iodoarenes as well as heterocycles delivered the corresponding products in excellent yields (**4a-4w**). Alkyl iodide was also competent, the unsymmetrical dialkylselenide (**4x-4z**) was obtained, although elevated temperature was required to achieve satisfactory yield. The methodology is also amenable to the utilization of different acrylamides as acceptors (Fig. 3b). Both electron-donating and electron-withdrawing substituents at *para-* and *meta-*position on the aryl group bound to nitrogen were well tolerated (**5-7**). N-alkylacrylamides (**8** and **9**) as well as (2E)-N-phenyl-butenamide (**10**) were also compatible, although the reactivity is moderate. Moreover, our protocol could be further applied to the facile synthesis of unsymmetrical monosulfides and monotellurides in moderate to good yield (**11a-11n** and **12a-12n**, Fig. 3c and 3d).

Having demonstrated the validity of our method, we next used our solid-state strategy to target other selenation processes that involved alternative electrophilic traps (Fig. 4a). Nucleophilic substitution of 2-chloropyrimidine (**13**) and 2-fluoroacetophenone (**20**) afforded diarylselenoethers (**14** and **21**) efficiently. In addition, alkyl iodides were also compatible, providing versatile alkyl-heteroaryl monoselenides (**15-19**) in good yield. As an alternative, switching the electrophile with alkyl triflate (**24**) and alkyl iodide (**27**) provided corresponding dialkylselenothers (**26** and **28**) with excellent yield.

products were isolated in satisfying results (**2n** and **2o**, 88% and 73%). Substrates with relatively strong electron-withdrawing nitrile substitution at *ortho-* or *para-* position (**1l** and **1i**) showed decreased reactivity while increasing the loading of magnesium as well as the vibration frequency significantly improved the yields (**2l** and **2i**, 68% and 74%). It is worth noting that both elemental sulfur and tellurium were also applicable, providing symmetrical disulfides and ditellurides

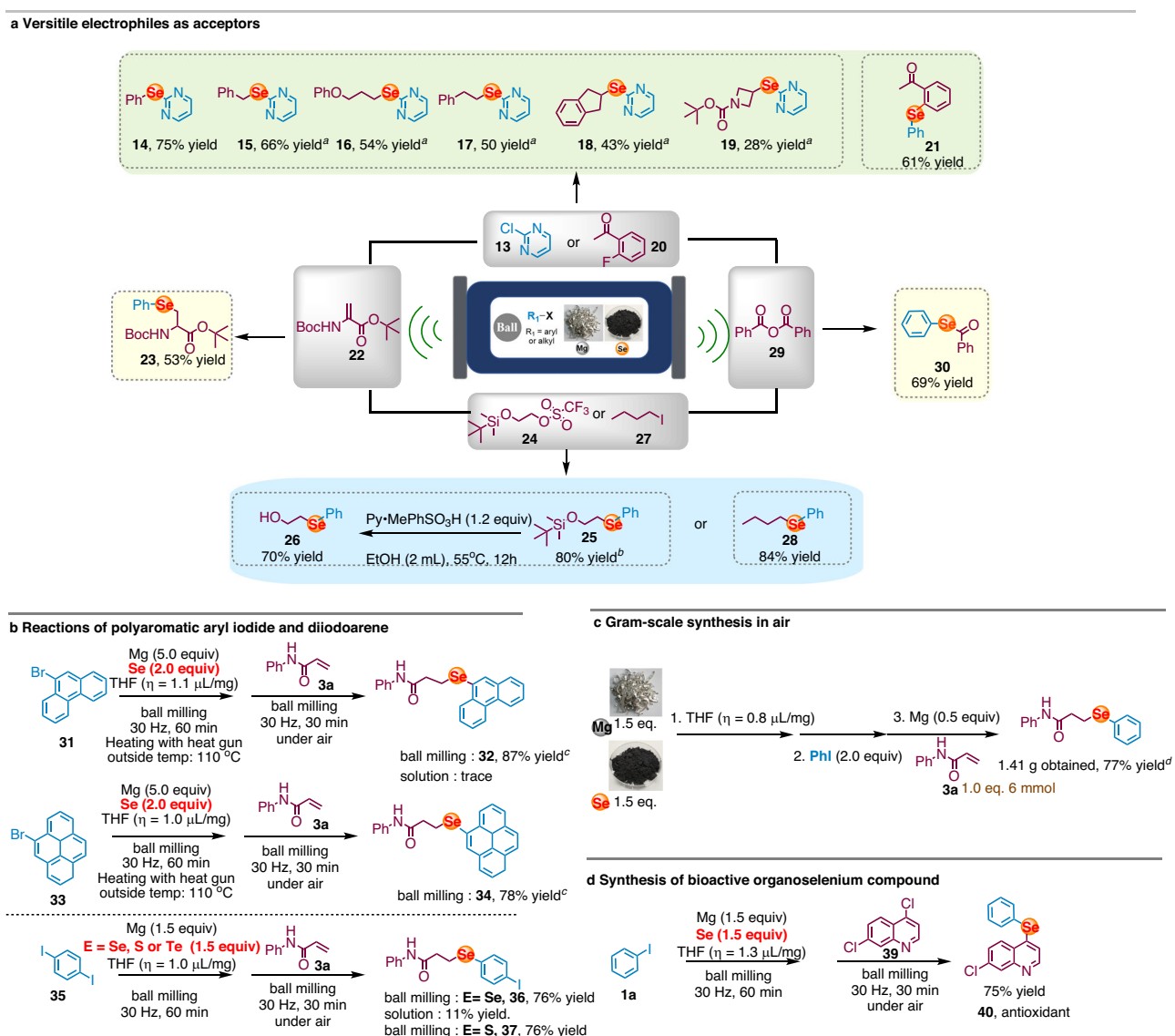

**Fig. 4 | Solid-state selenations using challenging substrates, their application in scale-up and bioactive antioxidant synthesis. a** Versatile electrophiles as acceptors; **b** Reactions of polyaromatic aryl iodide and diiodoarene; **c** Gram-scale synthesis in air; **d** Synthesis of bioactive organoselenium compound. Reaction conditions: a stainless-steel milling jar (1.5 mL) and a stainless-steel ball (diameter: 6 mm) were used; isolated yields. For details, see the Supplementary Information. [a] Additional Mg (1.0 equiv) was added in the second step. For details, see Supplementary Information Condition J. [b] **24** (1.0 mmol), Mg (0.5 equiv), Se (0.5 equiv), **1a** (0.5 equiv), LiCl (0.75 equiv), THF ($\eta$ = 0.8 μL/mg) in stainless-steel milling jar (5 mL) with a stainless-steel ball (diameter: 10 mm). For details, see Supplementary Information Condition I. [c] **3a** (0.5 mmol), Mg (5.0 equiv), Se (2.0 equiv), **31/33** (2.0 equiv), THF ($\eta$ = 1.1 μL/mg) in stainless-steel milling jar (5 mL) with a stainless-steel ball (diameter: 10 mm). For details, see Supplementary Information Condition G. [d] **3a** (6 mmol), THF ($\eta$ = 0.8 μL/mg) in stainless-steel milling jar (10 mL) with a stainless-steel ball (diameter: 15 mm). For details, see the Supplementary Information Methods' synthesis of unsymmetrical monochalcogenides with gram-scale reaction.

Conjugate addition reaction using α, β-unsaturated amino acid derivative (**22**) as electrophile provided selenium-containing amino acid in good yield (**23**). The protocol could also be applied to the facile construction of selenoesters, a type of important synthetic intermediates and widely applied to the synthesis of important bioactive compounds[74–78]. Using anhydride (**29**) as trapping reagent, corresponding selenol ester (**30**) was synthesized with significantly improved step and atomic efficiency compared with other strategies[79–82].

Large polyaromatic aryl halides (**31** and **33**) were submitted to the one-pot selenation sequence using *N*-phenylacrylamide as electrophile. Previous studies have suggested that external heating can improve the mixing efficiency and promote chemical reactions in solid state[83–85]. To further enhance reactivity, we employed a commercially

available, temperature-controllable heat gun, positioned it directly above the ball-milling jar, with the temperature set at 110 °C (For detailed screening of reaction parameters, see the Supplementary Information, Supplementary Fig. 4.). While solution-based condition failed, our strategy performed effectively under the optimized condition, yielding selenation product **32** and **34** in good yields. In the case of diiodoarene (**35**), monoselective selenation in solid state afforded **36** in good yield and excellent selectivity, while solution-based condition provided complex crude mixture with only 10% yield of product. Furthermore, this reaction offers the advantage of yielding mono-chalcogenides with good yields and selectivity (**37** and **38**). The robust nature of the current process was further highlighted by scaled-up experiment using bench THF as LAG in air (Fig. 4c) with slightly modified procedure, which successfully produced desired product

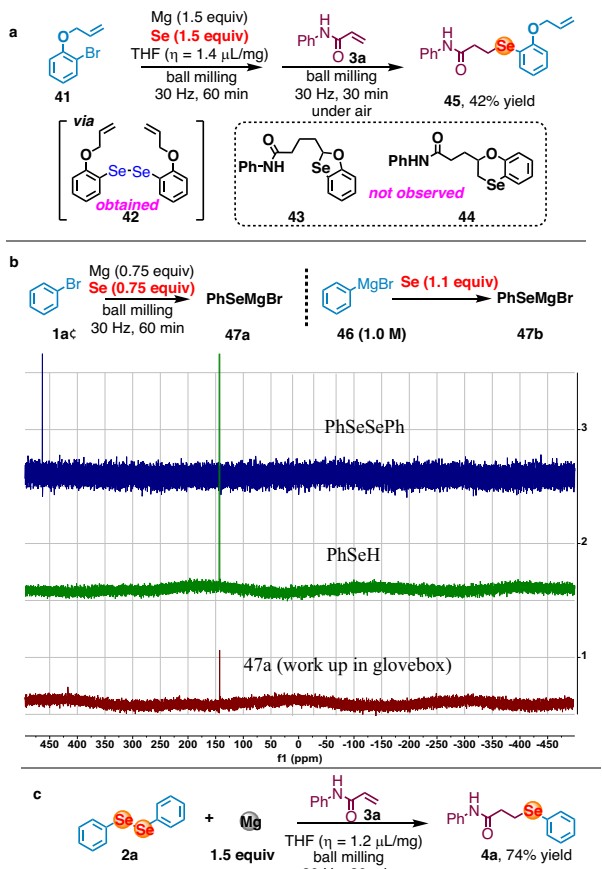

**Fig. 5 | Mechanistic Insights. a** Radical cyclization experiment. **b** NMR of intermediate. **c** Cleavage of Se-Se bond using magnesium.

without using anhydrous solvent as well as tedious schlenk technique. Moreover, our method could also be applied to the facile synthesis of bioactive compounds. Starting from 4,7-dichloroquinoline (**39**), **40** with potential antioxidant and anticholinesterase activities could be efficiently synthesized regioselectively in one-pot with 75% yield[86].

Preliminary mechanistic experiments were carried out to shed light on the ball-milling-enabled one-pot selenation process. Substrate with radical cyclization potential (**41**) was evaluated and only linear symmetrical diselenide intermediate (**42**) and unsymmetrical monoselenide (**45**) was obtained exclusively without observation of any cyclized byproduct (**43** or **44**), which indicated the reaction may proceed via two electron 1,4- nucleophilic addition pathway while not radical pathway initiated by one electron reduction of diselenide from magnesium under ball-milling condition[87] (Fig. 5a). Conducting standard reaction under argon protection and subsequently performing the work-up via short column filtration in a glove box resulted in the exclusive formation of benzeneselenol (as depicted in Fig. 5b, **47a**). In contrast, in previous experiments depicted in Fig. 2, when the work-up was conducted in the presence of air, it exclusively led to the formation of the symmetrical diselenide. This outcome strongly indicates that the Se-Se bond forming process occurs via air oxidation. Moreover, in the presence of magnesium, diphenyl diselenide (**2a**) could be converted to corresponding product **4a** efficiently within 30 min (Fig. 5c). It is worth noting that the cleavage of Se–Se bond using magnesium alone was previously regarded as a challenging task[88]. This observation indicates that symmetrical diselenides, which are generated in situ upon exposure to air, can be efficiently converted to selenium nucleophiles in the presence of magnesium under ball-milling condition. Subsequent exposure to electrophiles led to the formation of unsymmetrical monoselenide in good yields.

To confirm the generation of magnesium-based selenium nucleophiles under ball-milling conditions, we utilized Near-edge X-ray absorption fine structure (NEXAFS) spectroscopy. NEXAFS measurements were conducted at BL08U1A beamline of Shanghai Synchrotron Radiation Facility (SSRF in China) using magnesium-based selenium nucleophiles **47a** which were prepared through ball milling and then transferred into the soft X-ray optics under an argon atmosphere. The formation of the divalent cationic $Mg^{2+}$ species was unequivocally confirmed through the high-energy shift of the Mg K-absorption edge (1317.0 eV) in comparison to the $Mg^0$ edge (1315.0 eV) of a standard magnesium flake[89].

(Figure 6a). The high resemblance of the NEXAFS spectra at Mg-K, C-K Se-$L_3$ edges between the mechanochemically-prepared **47a** and PhSeMgBr prepared in solution **47b**[90] (Fig. 5b) supports the formation of similar magnesium-based organoselenium species with carbon-selenium bonds under both ball-milling and solution conditions in THF (Fig. 6a–c) The presence of carbon-selenium bond, arising from the transformation of the C-Br bond in the starting bromobenzene, was supported by the intense 1s–π* transition peaks at approximately 284.0 and 286.2 eV in the C-K NEXAFS spectra (Fig. 6b)[91]. Additionally, the formation of the monovalent anionic Se⁻ species was conclusively confirmed through the low-energy shift of the Se $L_3$-edge absorption peak (1445.3 eV) in **47a** relative to the $Se^0$ peak (1446.0 eV) in standard selenium powder (Fig. 6c)[92]. The remarkable similarity of Se $L_3$-edge, Mg K-edge and C K-edge NEXAFS spectra of mechanochemically-prepared **47a** to those of PhSeMgBr (**47b**) prepared in solution (Fig. 6a-c) further supports the formation of similar organoselenium species with carbon-selenium bonds under both ball-milling and solution conditions in THF[60].

## Discussion

In summary, we developed the ball-milling-enabled C-Se bond formation from readily accessible organic halides, magnesium metal and elemental selenium. The reaction features a wide substrate scope, tolerating protonic, steric and electronic different functionalities. The simple and efficient one-pot operation afforded a wide range of symmetrical diselenides and unsymmetrical monoselenides, respectively, whose formation would otherwise require transition metal catalysts, large excess of hazardous organic solvents and extensive heating. The synthetic potential of the solid-state selenation protocols was showcased through the facile access to a selenium-containing biologically important molecule (**40**). We envisage that our solid-state one-pot selenation strategy will serve as a launchpad for the invention of chalcogenation processes and related projects are ongoing in our laboratory.

## Methods

### General procedure for the synthesis of symmetrical dichalcogenides

Mg turnings (0.2 mmol, 1.0 equiv, 4.8 mg) and Se powder (0.2 mmol, 1.0 equiv, 16.0 mg) were placed in a jar (stainless-steel; 1.5 mL) with a ball stainless-steel; 6 mm, diameter) in argon. An organic halide **1** (0.2 mmol, 1.0 equiv), LiCl (0.3 mmol, 1.5 equiv, 12.7 mg) and THF (1.1-1.6 μL/mg) were added to the jar using a syringe. After the jar was closed, the jar was placed in a ball mill (Retsch MM 400, 15–30 min, 30 Hz). After grinding for 15–30 min, the mixture was eluted from silica gel with EA (ethyl acetate), the solvent was removed by vacuum distillation, and the pure product was obtained by rapid column chromatography (SiO₂, Hexane).

### General procedure for the synthesis of unsymmetrical monochalcogenides

Mg turnings (0.3 mmol, 1.5 equiv, 7.2 mg) and Se powder (0.3 mmol, 1.5 equiv, 24.0 mg) were placed in a jar (stainless-steel; 1.5 mL) with a ball (stainless-steel; 6 mm, diameter) in argon. An organic halide **1**

### a Mg K-edge NEXAFS spectra of 47a and 47b

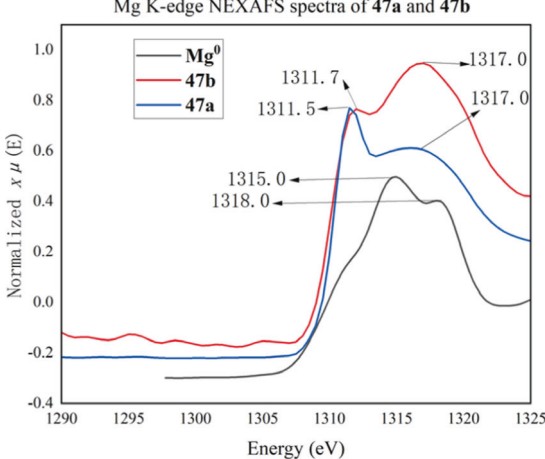

### b C K-edge NEXAFS spectra of 47a and 47b

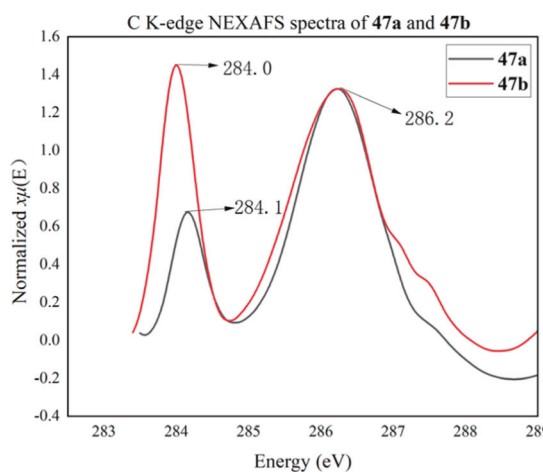

### c Se L₃-edge NEXAFS spectra of 47a and 47b

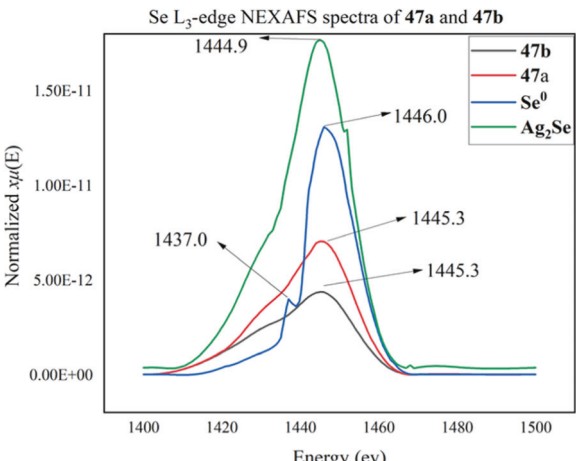

**Fig. 6 | NEXAFS analysis of a magnesium-based selenium nucleophiles under mechanochemical conditions. a** Mg K-edge NEXAFS spectra of mechanochemically prepared **47a, 47b**, and a magnesium flake of Mg⁰ standard. **b** C K-edge NEXAFS spectra of mechanochemically prepared **47a, 47b**. **c** Se L₃-edge NEXAFS spectra of mechanochemically prepared **47a, 47b** and selenium flake of Se⁰ and Ag₂Se standard.

(0.4 mmol, 2.0 equiv) and THF (0.8-1.4 µL/mg) were added to the jar using a syringe. After the jar was closed, the jar was placed in a ball mill (Retsch MM 400, 60 min, 30 Hz). After grinding for 60 min, the jar was opened in air and charged with an electrophile **3** (0.20 mmol, 29.4 mg). The jar was then closed without purging with inert gas, and was placed in the ball mill (Retsch MM 400, 30-60 min, 30 Hz). After grinding for 30-60 min, the mixture was eluted from silica gel with EA (ethyl acetate), the solvent was removed by vacuum distillation, and the pure product was obtained by rapid column chromatography (SiO₂, petroleum ether/ethyl acetate, 10:1-5:1).

## Data availability

For full characterization data including NMR/IR/HR-MS spectra of the new compounds and experimental details, see the Supplementary Material. All relevant data underlying the results of this study are available from the corresponding authors upon request.

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

## Acknowledgements

We thank the High-Level Introduction Plan of Shaanxi Province (71240000000111, X. W.), Natural Science Foundation of Shaanxi Province (2022JM-085, X. W.; 2022JQ-121, C. P.), the start-up funds from Xi'an Jiaotong University (X. W.), National Natural Science Foundation of China (22101224, C. F.), the China Postdoctoral Science Foundation (2021M692544, C. F.), the Fundamental Research Funds for the Central Universities (xzy012021049, C. F.) and Young Talent Fund of Xi'an Association for Science and Technology (095920221321, C. F.). We thank Dr. C. Feng and Dr. G. Chang from the Instrument Analysis Center of XJTU for assistance with NMR analysis. We are grateful to Prof. Rui Shang at the University of Tokyo and Prof. Yohei Shimizu at Hokkaido University for the helpful discussion. We also thank BL08U1A beamline of Shanghai Synchrotron Radiation Facility (SSRF) for providing beamtime.

## Author contributions

S. C., C. P. and X. W. conceived the study and designed the experiments. S. C., M. P., J. W., Y. Z. and Z. X. performed the experiments, mechanistic studies, and analyzed the data. J. Z., Jiyu Li., Junliang Lu, C. P. and X. W. directed the project. C. F. and Z. X. made contributions during the revision. C. P., C. F. and X. W. wrote the manuscript with input from all the other authors. All authors discussed the results and the manuscript.

## Competing interests

The authors declare no competing interests.
