## [Peer Review File · Nature Communications]

Reviewers' Comments:

Reviewer #1:

Remarks to the Author:

The paper presents the synthesis of various selenides via liquid assisted grinding (LAG). The methods are useful, but in my estimation represent a fairly modest step forward in methodology that likely does not meet the expectations of N. Commun.

In my reading of the manuscript, the key advance is the discovery that aryl diselenides can be formed via the LAG of elemental selenium and various aryl halides. The halides are typically fairly reactive (pyrimidine chloride, iodides, bromides, etc...) but the overall scope demonstrated is reasonably broad.

The subsequent chemistry seems to be highly precedented. I do not have a long personal history with selenium chemistry, but the examples reported here seem like natural extensions of known diselenide mechanochemistry, for example those presented in ref. 34. Essentially, the advance is that this known mechanochemical reactant can be generated in situ as an intermediate in a single pot. Certainly this is useful, and the observation is not obvious, but I am not convinced in the current paper that the novelty and impact is appropriate for this journal. From a mechanochemical reactivity point of view, the result is empirical, and so it is not clear that there are transferable mechanistic insights that might be applied to other systems. I would of course defer to those with more directly methodological experience that is relevant in the event that I underestimate the significance of the practical implications.

Whether considered further here or elsewhere, the manuscript would benefit from additional editing. The grammar starts well but becomes increasingly erroneous as the paper goes on. A few of these are very easily fixed. "Mordified" is used on multiple occasions in place of (I presume) "modified" (and not "mortified"). "Mehtod" is another typographical error, as is the missed boldface in 1r. The more problematic involve sentence structure, mismatches between subject and verb, verb tense, etc...

Figure 1a does not include examples of Se catalysis or Se based materials, as promised in the text.

I might be missing it, but the top of Figure 4a does not seem to show all reactants required for formation of the products shown.

Reviewer #2:

Remarks to the Author:

In this paper, Peng, Wei, and their colleagues present a novel approach for synthesizing a range of organoselenium compounds using ball-milling activation of elemental selenium. By utilizing this method, the authors demonstrate the successful synthesis of symmetrical diselenides and unsymmetrical monoselenides, offering valuable insights into the reactivity of selenium under ball-milling conditions. Additionally, the authors showcase gram-scale synthesis in air and the regioselective addition for the synthesis of bioactive organoselenium compounds. Mechanistic studies are also presented to shed light on the underlying reactions. The paper exhibits significant scientific novelty within the field of mechanochemistry and holds promise for practical applications. However, before publication in Nature Communications, several crucial points must be addressed.

Introduction: The authors should include a discussion on pioneering studies related to the mechanochemical synthesis of Grignard reagents, as this topic is highly relevant to the present research.

Ball-milling process: To provide clarity, the authors should elaborate on whether both magnesium and elemental selenium are activated by the ball-milling process. Furthermore, a more detailed

explanation of the working hypothesis would greatly benefit the readers.

Solvent amount: While the synthetic utility is evident, the use of 12 equivalents of THF, which can be regarded as a solvent amount raises concerns regarding the environmental sustainability and greenness of the strategy.

Spectroscopic evidence: It is essential to obtain spectroscopic evidence confirming the generation of Ar-Se-MgX under ball-milling conditions. Additionally, the existence of a reaction equilibrium between symmetrical diselenides and magnesium-based selenium nucleophiles requires substantiation. The authors should propose a mechanism for this equilibrium.

Mechanistic studies: Although the authors provide preliminary mechanistic studies, a conclusive summary or overall reaction pathway is currently absent. The authors should aim to provide a comprehensive understanding of the reaction mechanism.

Poorly soluble substrates: While the authors investigated the reaction of the polyaromatic halide 31, it is not considered a poorly soluble compound. To effectively demonstrate the utility of the mechanochemical protocol for reactions with poorly soluble substrates, the authors should investigate a broader range of large polyaromatic halides.

Missing references: The following references, which are highly relevant to the topic, should be included:

- a. doi.org/10.1038/s44160-022-00106-4
- b. doi.org/10.1002/anie.202217723
- c. doi.org/10.1039/D2SC05468J
- d. doi.org/10.1002/anie.202207118

Addressing these points will significantly enhance the quality and impact of the paper, making it suitable for publication in Nature Communications.

Reviewer #3:

Remarks to the Author:

Cheng Peng, Xiaofeng Wei et al. describe a strategy for the synthesis of organoselenium compounds under mild conditions using magnesium-based selenium nucleophiles. These species were formed in situ by milling organic halides, magnesium metal, and elemental selenium under liquid-assisted grinding (LAG) conditions. The method developed, was tested and used to produce symmetrical and asymmetric diselenides. In addition, the methodology was extended to sulfur and tellurium.

The manuscripts methodology seems broadly applicable and versatile, therefore it could potentially be of interest to the broad readership of Nature Communications. However, before the paper could be fully considered for publication there are several questions that need to be addressed.

The authors claim that sulfur is inert to ball milling. However, chalcogenides have been successfully used in mechanochemical reactions. I would suggest indicating that it has been previously used and adding the following citations.

- <https://doi.org/10.1002/ejic.201701414>;
- <https://doi.org/10.1021/acs.iecr.1c03784>;
- <https://doi.org/10.1039/C8CC01043A>;
- <https://doi.org/10.1039/C6CP03004A>

the authors do not properly compare the reaction between Se (the majority of the substrate scope)

and the S and Te counterparts. This should be added for a better assessment of the generality of the approach.

The authors claim to be the first reaction to form C-E bonds in ball milling. However, they do not cite <https://doi.org/10.1021/jo402071b>, where a series of S, Se and Te compounds are prepared using R2E as a source of chalcogen. I acknowledge that there are differences between the reagents used in their study and the published work. However the formation of these bonds have been previously been reported and needs to be indicated in the revised version of the manuscript/

Throughout the manuscript, the authors indicate the use of LAG. However, there is an apparent lack of calculations for "eta" values (microL/mg), which is common practice. The authors only indicate 10 equivalents of solvent. Please calculate (and clearly indicate throughout the manuscript) eta values and the rheology of the mixture, since it gives the impression of an almost dry (solvent-free) milling approach, but in reality, due to the use of 10 equivalents of THF, it might be close to being a slurry.

Despite having pictures on the ESI, the authors do not mention heating during the manuscript. This approach should be mentioned, discussed and cited. For instance, work using heating mantles in organic synthesis is not cited.

<https://doi.org/10.1038/s41570-022-00442-1>

<https://doi.org/10.1038/s41467-021-25495-6>

<https://doi.org/10.1002/ange.202007815>

Please check the English to correct typos (mordified, compitable, etc.).

In the ESI, please assign the peakf in the NMR spectra.

Responses to the Reviewers' Comments

We thank the reviewer for his/her valuable suggestion and comments.

Dear Editors and Reviewers,

We appreciate the efforts you have made to evaluate our work. Thank you for giving us an opportunity to improve our manuscript entitled "Mechanochemical Synthesis of Organoselenium Compounds" (**Manuscript Number: NCOMMS-23-22030**). We appreciate your valuable comments, which are significantly helpful for us to improve the quality of our manuscript. We have addressed all issues in positive manners, and have revised the manuscript accordingly with new data and their interpretation added. The revised manuscript and marked version with highlighted changes are now resubmitted for your evaluation. Please find our point-by-point responses provided below.

Reviewer: 1

The paper presents the synthesis of various selenides via liquid assisted grinding (LAG). The methods are useful, but in my estimation represent a fairly modest step forward in methodology that likely does not meet the expectations of N. Commun.

In my reading of the manuscript, the key advance is the discovery that aryl diselenides can be formed via the LAG of elemental selenium and various aryl halides. The halides are typically fairly reactive (pyrimidine chloride, iodides, bromides, etc...) but the overall scope demonstrated is reasonably broad.

Response: We express our gratitude to the reviewer for recognizing the key advance of our discovery and commented the scope of our method as "reasonably broad". In chalcogenation chemistry, direct conversion of elemental chalcogen into valuable organochalcogenides is impressively appealing as the method obviates the need of sensitive and toxic selenium reagents. Mechanochemistry, an emerging field in organic chemistry, holds the for efficient and user-friendly transformation handling chalcogen elements.

In this study, our primary objective was to establish a highly practical platform for the comprehensive synthesis of organochalcogen compounds. Our scope extends beyond "fairly reactive aryl halides" and encompasses heteroaryl-, electron-withdrawing or electron-donating aryl halides, as well as alkyl halides. This versatile transformation includes conjugate addition, substitution reactions, and transesterification. Remarkably, the reaction can accommodate sensitive protic functional groups, leading to a diverse array of structures with potential applications in bioactivity evaluation.

The subsequent chemistry seems to be highly precedented. I do not have a long personal history with selenium chemistry, but the examples reported here seem like natural extensions of known diselenide mechanochemistry, for example those presented in ref. 34. Essentially, the advance is that this known mechanochemical reactant can be generated in situ as an intermediate in a single pot. Certainly this is useful, and the observation is not obvious, but I am not convinced in the current paper that the novelty and impact is appropriate for this journal. From a mechanochemical reactivity point of

view, the result is empirical, and so it is not clear that there are transferable mechanistic insights that might be applied to other systems. I would of course defer to those with more directly methodological experience that is relevant in the event that I underestimate the significance of the practical implications.

Response: We appreciate the critical analysis of from this reviewer. We were also encouraged to see the reviewer recognized the significance of the practical implications. We also provided additional mechanistic investigation and interpretation in our revised manuscript. This work represents the first instance of a mechanochemical carbon-chalcogen bond-forming reaction using elemental chalcogen. It offers one of the most straightforward methods for the systematic synthesis of organochalcogenides. Conventional preformed activated selenium reagents, such as known diselenides, encounter several issues, including relatively high cost, storage challenges (especially concerning the unpleasant smell of alkyl diselenides), and limited commercial availability, among others. As demonstrated in the main text, current methods often rely on the use of expensive transition metals and involve the utilization of large quantities of toxic polar organic solvents. Our current strategy has the potential to address these problems effectively.

Throughout the revision process, we conducted extensive investigations into the application scope of the current strategy. This method has demonstrated its capability to efficiently produce a wide range of organochalcogenides with high yields, step-efficiency, and atom-economy. The analysis using NEXAFS spectroscopy, along with preliminary mechanistic investigations, suggests the strategic potential that can combine magnesium-based organochalcogen reagents as coupling partners

Whether considered further here or elsewhere, the manuscript would benefit from additional editing. The grammar starts well but becomes increasingly erroneous as the paper goes on. A few of these are very easily fixed. “Mordified” is used on multiple occasions in place of (I presume) “modified” (and not “mortified”). “Mehtod” is another typographical error, as is the missed boldface in 1r. The more problematic involve sentence structure, mismatches between subject and verb, verb tense, etc...

Response: We sincerely apologize for numerous grammatic errors that may have caused inconvenience to the reviewer during the evaluation of our manuscript. We appreciate the meticulous work of this reviewer to point the grammar and spelling errors. We comprehensively checked and modified the text by consulting a native speaker. The corresponding changes could be found in the text with highlighted mark.

Figure 1a does not include examples of Se catalysis or Se based materials, as promised in the text.

Response: Thank you for the comment. We have updated the information in Fig 1a accordingly.

I might be missing it, but the top of Figure 4a does not seem to show all reactants

required for formation of the products shown.

Response: Thank you for the comment. We revised Fig. 4a accordingly to include all the required information in our revised manuscript.

Reviewer: 2

In this paper, Peng, Wei, and their colleagues present a novel approach for synthesizing a range of organoselenium compounds using ball-milling activation of elemental selenium. By utilizing this method, the authors demonstrate the successful synthesis of symmetrical diselenides and unsymmetrical monoselenides, offering valuable insights into the reactivity of selenium under ball-milling conditions. Additionally, the authors showcase gram-scale synthesis in air and the regioselective addition for the synthesis of bioactive organoselenium compounds. Mechanistic studies are also presented to shed light on the underlying reactions. The paper exhibits significant scientific novelty within the field of mechanochemistry and holds promise for practical applications.

However, before publication in Nature Communications, several crucial points must be addressed.

Response: We thank this reviewer for the very positive evaluation and his/her support for publishing our work in Nature Communication.

Introduction: The authors should include a discussion on pioneering studies related to the mechanochemical synthesis of Grignard reagents, as this topic is highly relevant to the present research.

Response: We greatly appreciated the reviewer's suggestion. Accordingly, we have added studies related to the mechanochemical synthesis of Grignard reagents to the revised manuscript.

In page 1, line 44-53, we have added “ Recently, the strategy has been applied to facilitate the oxidative addition of zero valent metals (such as Mg, Mn, Zn, Ca, etc) to organic halides, generating organometallic species for diverse nucleophilic transformations.⁴⁸⁻⁶⁰ For instance, Ito, Kubota,⁶⁰ and Bolm⁶¹ groups independently demonstrated elegant examples of producing versatile Grignard reagents using mechanochemical strategy. Remarkably, in the work of Ito and Kubota, ⁶⁰ where the milling jar was briefly exposed to the air before adding an electrophile, no degradation of air-sensitive Grignard reagents was observed, highlighting the method's robustness.”. In “Fig 1. Synthesis of organoselenium compounds”, we have added “c) The mechanical activation of zero-valent metals”.

Ball-milling process: To provide clarity, the authors should elaborate on whether both magnesium and elemental selenium are activated by the ball-milling process. Furthermore, a more detailed explanation of the working hypothesis would greatly benefit the readers.

Response: We sincerely thank the reviewer for the valuable suggestion, and in response,

we have performed Near-edge X-ray absorption fine structure (NEXAFS) spectroscopy experiments to observe the formation of intermediates and we have subsequently incorporated the additional results and analysis into the main text.

In page 5, line 3-10 and line 16-39, we have added “To confirm the generation of magnesium-based selenium nucleophiles under ball-milling conditions, we utilized Near-edge X-ray absorption fine structure (NEXAFS) spectroscopy. NEXAFS measurements were conducted at BL08U1A beamline of Shanghai Synchrotron Radiation Facility (SSRF in China) using magnesium-based selenium nucleophiles **47a**, which were prepared through ball milling and then transferred into the soft X-ray optics under an argon atmosphere. The formation of the divalent cationic Mg^{2+} species was unequivocally confirmed through the high-energy shift of the Mg K-absorption edge (1317.0 eV) in comparison to the Mg^0 edge (1315.0 eV) of a standard magnesium flake (Fig. 6a).⁸⁹ The high resemblance of the NEXAFS spectra at Mg-K, C-K Se-L₃ edges between the mechanochemically-prepared **47a** and PhSeMgBr prepared in solution **47b**⁹⁰ (Fig. 5b) supports the formation of similar magnesium-based organoselenium species with carbon–selenium bonds under both ball-milling and solution conditions in THF (Fig. 6a-c). The presence of carbon-selenium bond, arising from the transformation of the C-Br bond in the starting bromobenzene, was supported by the intense $1s-\pi^*$ transition peaks at approximately 284.0 and 286.2 eV in the C-K NEXAFS spectra (Fig. 6b).⁹¹ Additionally, the formation of the monovalent anionic Se^{1-} species was conclusively confirmed through the low-energy shift of the Se L₃-edge absorption peak (1445.3 eV) in **47a** relative to the Se^0 peak (1446.0 eV) in standard selenium powder (Fig. 6c).⁹² The remarkable similarity of Se L₃-edge, Mg K-edge and C K-edge NEXAFS spectra of mechanochemically-prepared **47a** to those of PhSeMgBr (**47b**) prepared in solution (Fig. 6a-c) further supports the formation of similar organoselenium species with carbon–selenium bonds under both ball-milling and solution conditions in THF.⁶⁰”

In page 5, we have added “Fig 6. NEXAFS analysis of a magnesium-based selenium nucleophiles under mechanochemical conditions.”

a. NEXAFS analysis of 47a prepared under mechanochemical conditions

b. Mg K-edge NEXAFS spectra of 47a and 47b

c. C K-edge NEXAFS spectra of 47a and 47b

d. Se L₃-edge NEXAFS spectra of 47a and 47b

Fig 6. NEXAFS analysis of a magnesium-based selenium nucleophiles under mechanochemical conditions.

References

89. Witte, K. et al. NEXAFS spectroscopy of chlorophyll a in solution. *J. Phys. Chem. B* 120, 11619-11627 (2016)
90. Cooney, R. R. & Urquhart, S. G. Chemical trends in the near-edge X-ray absorption fine structure of monosubstituted and para-bisubstituted benzenes. *J. Phys. Chem. B* 108, 18185-18191 (2004).
91. Derfus, A. M., Chan, W.C.W., Bhatia, S.N. Probing the Cytotoxicity of Semiconductor Quantum Dots. *Nano Lett.* 4, 11-18 (2004).
92. Li Y, et al. Degradable Selenium-Containing Polymers for Low Cytotoxic Antibacterial Materials. *ACS Macro Lett.* 11, 1349-1354 (2022).
60. Takahashi, R., et al. Mechanochemical synthesis of magnesium-based carbon nucleophiles in air and their use in organic synthesis. *Nature Commun.* 12, 6691 (2021).

In page 1, line 53-76, we described the working hypothesis as the following“In this report, we present a mechanochemical method for synthesizing organoselenium compounds which involves the in situ generation of magnesium-based selenium species through the straightforward process of mixing and grinding organic halides, magnesium, and elemental selenium. Notably, these species exhibit extreme sensitivity to both oxygen and water, leading to their complete conversion into symmetrical diselenides during the work up procedure. Additionally, employing a one-pot process for the addition of electrophiles enabled efficient synthesis of unsymmetrical monoselenides, which proceeded smoothly even in the presence of air. (Fig 1d). We also achieved the successful preparation of magnesium-based organoselenium reagents from polyaromatic aryl halides and diiodoarenes. Notably, it's important to highlight that converting such substrates into organochalcogenides poses challenges when employing conventional solution-based methods. Near edge, X-ray absorption fine structure (NEXAFS) spectroscopy was employed to analyze the generation of the magnesium-based organoselenium nucleophiles under mechanochemical conditions. The method can be extended to the straightforward synthesis of organoic sulfur and tellurium compounds, suggesting its potential potential to serve as a highly practically foundation for the comprehensive synthesis of organochalcogen compounds.”

Solvent amount: While the synthetic utility is evident, the use of 12 equivalents of THF, which can be regarded as a solvent amount raises concerns regarding the environmental sustainability and greenness of the strategy.

Response: To better distinguish the amount of solvent in solution-based condition, we have changed the amount of LAG from “THF (xx eq.)” to “xx μ L/mg” in the revised manuscript and Supplementary Information.

In light of our commitment to environmental sustainability and green synthetic strategies, we have reduced the use of Liquid-Assisted Grinding (LAG) THF to 6.0 equivalents. This adjustment has been found to be equivalent in reactivity to using 12.0 equivalents of THF for the synthesis of dichalcogenides. As a result, we have set the amount of THF at 6.0 equivalents and re-evaluated the scope in Figure 2.

Corresponding condition screening details can be found in Supplementary Table 2. Furthermore, using the calculation formula for LAG ($\eta = V$ [liquid; μL]/ m [reagents; mg]), we calculated the " η " values ($\mu\text{L}/\text{mg}$) of symmetrical dichalcogenides within the range of 1.16-1.62 $\mu\text{L}/\text{mg}$. This demonstrates that the amount of THF we utilized aligns with solid-phase ball milling (0-2) $\mu\text{L}/\text{mg}$, as reported in *Adv. Synth. Catal.* **2021**, *363*, 1246-1271, and does not exceed the solvent amount.

Regarding the synthesis of unsymmetrical monochalcogenides, our Supplementary Information (Supplementary Table 3) shows that the reaction can still proceed with 3.0 equivalents of THF. With 6.0 equivalents of THF, the target compound can be obtained with a 76% yield. However, after careful consideration, taking into account factors such as the amount of THF, reaction time, and overall efficiency, we have determined that using 12.0 equivalents of THF achieves efficient synthesis with a 94% yield. Additionally, we calculated the " η " values ($\mu\text{L}/\text{mg}$) for unsymmetrical chalcogenides in the range of 0.81-1.88 $\mu\text{L}/\text{mg}$, which align with the mechanochemical definition of Liquid-Assisted Grinding (LAGs).

Spectroscopic evidence: It is essential to obtain spectroscopic evidence confirming the generation of Ar-Se-MgX under ball-milling conditions. Additionally, the existence of a reaction equilibrium between symmetrical diselenides and magnesium-based selenium nucleophiles requires substantiation. The authors should propose a mechanism for this equilibrium.

Response: We appreciate the insightful suggestion from this reviewer. As the answer of question 2, we provided spectroscopic evidence by using near-edge X-ray absorption fine structure (NEXAFS) spectroscopy, which support the generation of Ar-Se-MgX under ball-milling conditions.

The experiments conducted freshly, as shown in Fig 5b, have ruled out the existence of a reaction equilibrium between symmetrical diselenides and magnesium-based selenium nucleophiles. The complete formation of benzeneselenol upon quenching in the glovebox strongly suggests that the formation of symmetrical diselenide was indeed promoted by oxidation in the air during the work up process. As a result, we have made necessary modifications to the text on page 2, line 60-63, page 3, line 1-2 and page 4, line 24-32.

Mechanistic studies: Although the authors provide preliminary mechanistic studies, a conclusive summary or overall reaction pathway is currently absent. The authors should aim to provide a comprehensive understanding of the reaction mechanism.

Response: Based on preliminary mechanistic studies, as shown in Fig 5 with additional results and NEXAFS analysis presented in Fig 6, we propose the following reaction process, as shown in Fig 1d. Initially, under ball milling conditions, magnesium undergoes oxidative insertion into aryl halide to form **PhMgX** (*Nature Commun.*, **12**, 6691, (2021)), which subsequently reacts with elemental selenium to produce **PhSeMgX** (*J. Chem. Res.*, **34**, 127-129 (2010)). However, because we open the stainless-steel jar in the air and put **1a** in the stainless-steel jar, a portion of **PhSeMgX** is hydrolyzed to generate PhSeH due to the presence of moisture from the air (*Bioorg.*

Med. Chem. Lett., **29**, 126726(2019); *Chem. Eur.J.*, **23**, 2405 -2422(2017)) and it is oxidized by the air, forming symmetrical diselenides. Nevertheless, diselenides can be reconverted into selenium nucleophiles in the presence of excessive magnesium (Fig 5c), subsequently converting to unsymmetrical monoselenide in the presence of an electrophile.

Accordingly, the main text was modified to address the mechanism issue. As shown in Page 4, Line 17-40 and Page 5, Line 3-10, Line 16-39.

Poorly soluble substrates: While the authors investigated the reaction of the polyaromatic halide 31, it is not considered a poorly soluble compound. To effectively demonstrate the utility of the mechanochemical protocol for reactions with poorly soluble substrates, the authors should investigate a broader range of large polyaromatic halides.

Response: Thank you for the kind suggestion. We changed the content limited to large polyaromatic aryl halides without emphasizing poorly soluble compound. Moreover, we further explored the scope as shown in Figure 4b (**31** and **33**).

Missing references: The following references, which are highly relevant to the topic, should be included:

- a. doi.org/10.1038/s44160-022-00106-4
- b. doi.org/10.1002/anie.202217723
- c. doi.org/10.1039/D2SC05468J
- d. doi.org/10.1002/anie.202207118

Response: we apologize for missing these highly relevant references in our previous citations. According to the reviewer suggestion, we have cited all the four references in the revised manuscript.

In page 1, line 47, “(references 56) Gao, P., Jiang, J., Maeda, S., Kubota, K. & Ito, H. Mechanochemically Generated Calcium-Based Heavy Grignard Reagents and Their Application to Carbon-Carbon Bond-Forming Reactions. *Angew. Chem. Int. Ed.* **61**, e202207118 (2022).”

In page 1, line 47, “(references 57) Gao, Y., Kubota, K. & Ito, H. Mechanochemical Approach for Air-Tolerant and Extremely Fast Lithium-Based Birch Reductions in Minutes. *Angew. Chem. Int. Ed.* **62**, e202217723 (2023).”

In page 1, line 47, “(references 58) Takahashi, R., Gao, P., Kubota, K. & Ito, H. Mechanochemical protocol facilitates the generation of arylmanganese nucleophiles from unactivated manganese metal. *Chem. Sci.* **14**, 499-505 (2023).”

In page 1, line 47, “(references 59) Jones, A. C, Leitch, J.A, Raby-Buck. & S. E, Browne, D. L. Mechanochemical techniques for the activation and use of zero-valent metals in synthesis. *Nat. Synth.* **1**, 763-775 (2022).”

And In page 1, line 41-45, we have added these references to “Recently, the strategy has been applied to facilitate the oxidative addition of zero valent metals (such as Mg, Mn, Zn, Ca, etc) to organic halides, generating organometallic species for diverse nucleophilic transformations.⁴⁸⁻⁶⁰”. We have also added relevant references 56-59 to the References.

Reviewer: 3

Cheng Peng, Xiaofeng Wei et al. describe a strategy for the synthesis of organoselenium compounds under mild conditions using magnesium-based selenium nucleophiles. These species were formed in situ by milling organic halides, magnesium metal, and elemental selenium under liquid-assisted grinding (LAG) conditions. The method developed, was tested and used to produce symmetrical and asymmetric diselenides. In addition, the methodology was extended to sulfur and tellurium. The manuscripts methodology seems broadly applicable and versatile, therefore it could potentially be of interest to the broad readership of Nature Communications.

However, before the paper could be fully considered for publication there are several questions that need to be addressed.

Q 1: The authors claim that sulfur is inert to ball milling. However, chalcogenides have been successfully used in mechanochemical reactions. I would suggest indicating that it has been previously used and adding the following citations.

<https://doi.org/10.1002/ejic.201701414>;

<https://doi.org/10.1021/acs.iecr.1c03784>;

<https://doi.org/10.1039/C8CC01043A>;

<https://doi.org/10.1039/C6CP03004A>

Response: We thank this reviewer for the very positive evaluation and his/her support for publishing our work in Nature Communication.

We appreciate the valuable suggestions. We have cited the four references accordingly in the revised manuscript.

In page 2, line 35, “(references 70) Kumar, R., Kumar, S., Pandey, M. K., Kashid, V. S., Radhakrishna, L. & Balakrishna, M. S. Synthesis of phosphine chalcogenides under solvent-free conditions using a rotary ball mill. *Eur. J. Inorg. Chem.* **2018**, 1028-1037 (2018).”

In page 2, line 35, “(references 71) Sim, Y., Tan, D., Ganguly, R., Li, Y.-X. & García, F. Orthogonality in main group compounds: a direct one-step synthesis of air- and moisture-stable cyclophosphazanes by mechanochemistry. *ChemComm.* **54**, 6800-6803 (2018).”

In page 2, line 35, “(references 72) Chua, C. K., Sofer, Z., Khezri, B., Webster, R. D. & Pumera, M. Ball-milled sulfur-doped graphene materials contain metallic impurities originating from ball-milling apparatus: their influence on the catalytic properties. *Phys. Chem. Chem. Phys.* **18**, 17875-17880 (2016).”

In page 2, line 15-18, “(references 73) Geng, X.-Z., Zhao, W.-M., Zhou, Q., Duan, Y.-F., Huang, T.-F. & Liu, X.-S. Effect of the mechanochemical process on the stability of Mercury in simulated fly ash, part 2: sulfur additive. *Ind. Eng. Chem. Res.* **60**, 15115-15124 (2021).”

And In page 2, line 33-36, we have added these references to “Although investigation in chalcogenation process via ball-milling strategy is raising considerable concern,⁷⁰⁻⁷³ direct construction of carbon-chalcogen bond from chalcogen element is rare.” and added references 70-73 to the References.

The authors do not properly compare the reaction between Se (the majority of the

substrate scope) and the S and Te counterparts. This should be added for a better assessment of the generality of the approach.

Response: Follow the suggestions of the reviewer, we have added a total of 7 disulfides (**2rb-2rh** in Fig. 2b) and 7 ditelluriums (**2sb-2sh** in Fig.2c), 13 monosulfides (**11b-11n** in Fig. 3c) and 13 monotelluriums (**12b-12n** in Fig. 3c) in the revised manuscript.

In Page 2, line 26-69, we have added “It is worth noting that both elemental sulfur and tellurium were also applicable, providing symmetrical disulfides and ditellurides in moderate to good yields (**2ra-2rh** and **2sa-2sh**).” And in page 2, Fig 2b and Fig 2c. we have added “*b*) Elemental sulfur” and “*c*) Elemental telluride”

In page 3, line 33-36, We also added “Moreover, our protocol could be further applied to the facile synthesis of unsymmetrical monosulfides and monotellurides in moderate to good yield obtained corresponding product (**11a-11n** and **12a-12n**, Fig 3c and Fig 3d).”

In page 3, line 71-73 “Furthermore, this reaction offers the advantage of yielding monochalcogenides with exceptional yields and selectivity (**37** and **38**)” in the manuscript.

In page 3, Fig 3c and Fig 3d. we have added “*c*) Elemental sulfur” and “*d*) Elemental telluride”.

Through the application of this approach in synthesizing monosulfides and monotellurides, we were able to compare the reactivity of elemental sulfur, selenium, and tellurium more effectively. This further confirmed the broad applicability of our development strategy.

The authors claim to be the first reaction to form C-E bonds in ball milling. However, they do not cite <https://doi.org/10.1021/jo402071b>, where a series of S, Se and Te compounds are prepared using R₂E as a source of chalcogen. I acknowledge that there are differences between the reagents used in their study and the published work. However the formation of these bonds have been previously been reported and needs to be indicated in the revised version of the manuscript.

Response: We thank this reviewer for pointing out this important issue. To address this concern, we have restricted our description to “First ball-milling-enabled C-Se bond formation” in Page 5, 47-48 as well as “First mechanochemical C-Se bond formation from selenium” in Page 2, Fig 1d. Moreover, we have made necessary updates to the related references in the revised manuscript.

Throughout the manuscript, the authors indicate the use of LAG. However, there is an apparent lack of calculations for “eta” values (microL/mg), which is common practice. The authors only indicate 10 equivalents of solvent. Please calculate (and clearly indicate throughout the manuscript) eta values and the rheology of the mixture, since it gives the impression of an almost dry (solvent-free) milling approach, but in reality, due to the use of 10 equivalents of THF, it might be close to being a slurry.

Response: We appreciate this valuable suggestion. According to this suggestion, we

have calculated the “eta” values (microL/mg) in the revised manuscript and supplementary information, and changed the representation of LAG dosage from “THF (xx eq.)” to “xx $\mu\text{L}/\text{mg}$ ”.

First of all, our solid-liquid ratio ($\eta = V(\text{liquid}; \mu\text{L})/m(\text{reagents}; \text{mg})$) is 0.81-1.88 $\mu\text{L}/\text{mg}$, which is in the range of common liquid-solid ratio of ball milling (0-2 $\mu\text{L}/\text{mg}$) (*Adv. Synth. Catal.*, **363**, 1246 - 1271 (2021)). Secondly, THF as an activation factor of reaction can effectively improve the yield of large polyaromatic halide (**33**) under ball milling conditions. However, even with vigorous stirring conditions and heating under the same solution conditions, the product could not be obtained. Thus, we believe that the strategy is mechanochemical synthesis rather than slurry reaction.

A photo below shows the mixture in the jar after grinding and opening in the air to obtain the product (as shown):

Despite having pictures on the ESI, the authors do not mention heating during the manuscript. This approach should be mentioned, discussed and cited. For instance, work using heating mantles in organic synthesis is not cited.

<https://doi.org/10.1038/s41570-022-00442-1>

<https://doi.org/10.1038/s41467-021-25495-6>

<https://doi.org/10.1002/ange.202007815>

Response: We greatly appreciated the reviewer’s suggestion. We have added a description and discussion of the heating process in the revised manuscript, and cited relevant literatures (*Nat. Commun.* **12**, 6691 (2021); *J. Am. Chem. Soc.* **2021**, 143, 6165-6175, etc.).

In page 3, line 60, “(references 83) Yong. T., Bati. G., Garcıa. F. & Stuparu, M.C. Mechanochemical transformation of planar polyarenes to curved fused-ring systems. *Nat. Commun.*, **12**, 5187 (2021).”

In page 3, line 60, “(references 84) Martinez, V., Stolar, T., Karadeniz, B., Brekalo, I. & Uzarevic, K. Advancing mechanochemical synthesis by combining milling with different energy sources. *Nat. Rev. Chem.* **7**, 51-65 (2023).”

In page 3, line 60, “(references 85) Bati G, et al. Bati, G., et al. Mechanochemical synthesis of corannulene-based curved nanographenes. *Angew. Chem. Int. Ed.* **59**, 21620-21626 (2020).”

Meanwhile, the pictures of heating operations have been added to the revised supporting information (Supplementary Figure 4. Reaction set up with a heat gun). In Page 3, line 56-67, “Large polyaromatic aryl halide (**31**), generally considered as challenging substrate due to their limited solubility, was submitted to the one-pot selenation sequence using *N*-phenylacrylamide as electrophile. While solution-based condition failed, our ball-mill strategy performed well under the optimized condition to

provide selenation product **32** in good yields.” has been reviewed to “Large polyaromatic aryl halides (**31** and **33**) were submitted to the one-pot selenation sequence using *N*-phenylacrylamide as electrophile. Previous studies have suggested that external heating can improve the mixing efficiency and promote chemical reactions in solid state.^{60, 83-86} To further enhance reactivity, we employed a commercially available, temperature-controllable heat gun, positioning it directly above the ball-milling jar, with the temperature set at 110 °C (For detailed screening of reaction parameters, see the Supporting Information, Supplementary Figure 4.). While solution-based condition failed, our ball-mill strategy performed effectively under the optimized condition, yielding selenation product **32** and **34** in good yields.”

Supplementary Figure 4. Reaction set up with a heat gun

Please check the English to correct typos (mordified, compitable, etc.).

Response: We apologize for many grammatic errors and typos. We appreciate the meticulous work of this reviewer to check our manuscript. We comprehensively checked and modified the text by consulting a native speaker. The corresponding changes could be found in the text with highlight.

In the ESI, please assign the peak of in the NMR spectra.

Response: According to the reviewer’s suggestion, we have assigned the peaks of the NMR spectra (¹H NMR and ¹³C NMR), as detailed in the revised supporting information.

With the above changes and responses, I hope you will find this revised version suitable for publication in *Nature Communications*. Your kind consideration is highly appreciated.

Reviewers' Comments:

Reviewer #2:

Remarks to the Author:

The authors have revised manuscript according to the suggestions by the reviewers. The manuscript is now ready to publish.

Reviewer #3:

Remarks to the Author:

The revised version of the manuscript "Mechanochemical Synthesis of Organoselenium Compounds" by Xiaofeng Wei describes a new approach to synthesising organoselenium compounds under ball milling conditions.

The revised version addresses most of the reviewer's comments (references, expanding the discussion, etc.), including some mechanistic studies using NEXAFS - which strengthens their previous submission.

In the supporting information, the NMR figures are still not labelled. For each figure, please assign labels to each environment in each of the chemical structures and match them with the corresponding signal on the NMR spectra.

HR-MS and IR spectra figures should be included in the supporting information. The current ESI cites the values but does not include the spectra.

Some compounds in the ESI only include NMR values but do not include the IR and HR-MS data. These should be included if available. If not available, the authors should justify it.

Revising the manuscript, I noticed that the font size on the figures seems a bit small. Please check that the formatting complies with the recommended guidelines.

The data availability statement is too vague.

Responses to the Reviewers Comments

We thank the reviewer for his/her valuable suggestion and comments.

Dear Editors and Reviewers,

Thank you for giving us the opportunity in improving our manuscript entitled “Mechanochemical Synthesis of Organoselenium Compounds” (**Manuscript Number: NCOMMS-23-22030A**). We are very grateful to your highly valuable comments, which are significantly helpful to improve the quality of our manuscript. According to the feedbacks, we have carefully revised the manuscript with newly added data and interpretation, and resubmitted it for your evaluation. Revised portions are marked yellow in the manuscript. In addition, our point-by-point responses are provided below.

Reviewer: 3

The revised version of the manuscript “Mechanochemical Synthesis of Organoselenium Compounds” by Xiaofeng Wei describes a new approach to synthesising organoselenium compounds under ball milling conditions.

The revised version addresses most of the reviewer's comments (references, expanding the discussion, etc.), including some mechanistic studies using NEXAFS - which strengthens their previous submission

Q 1: In the supporting information, the NMR figures are still not labelled. For each figure, please assign labels to each environment in each of the chemical structures and match them with the corresponding signal on the NMR spectra.

A1: we greatly appreciated the reviewer's suggestion, and we have made the following modifications:

In Supplementary Information, page 57-292, top the correlated NMR spectrum, we added “¹H NMR (400 MHz or 600 MHz, CDCl₃, rt)/ ¹³C NMR (101 MHz or 150 MHz, CDCl₃, rt)/⁷⁷Se NMR (76 MHz, CDCl₃, rt) of compound Y” and added “Supplementary Fig. X. NMR spectra of compound Y” at the below of the NMR spectrum. Also, below the correlated IR and HR-MS spectrum, we also added “Supplementary Fig. X. IR spectra of compound Y” and “Supplementary Fig. X. HR-MS spectra of compound Y”.

Q 2: HR-MS and IR spectra figures should be included in the supporting information. The current ESI cites the values but does not include the spectra.

A2: we sincerely thank the reviewer for the valuable suggestion, and in response, we have added the correlated spectrum in the **Supplementary Information**, page 57-292.

Q 3: Some compounds in the ESI only include NMR values but do not include the IR

and HR-MS data. These should be included if available. If not available, the authors should justify it.

A3: Thanks for pointing out this question. For compounds that have already been reported, we tested the relevant ^1H NMR, ^{13}C NMR spectra and confirmed that they are consistent with the relevant references that have already been reported, and for compounds that have no reported ^{77}Se NMR, we supplemented the selenium profile data. For the new compounds that appear in our article, we measured NMR, IR, and HR-MS. For compound **25**, the error of the HR-MS test results is relatively large, so we have transformed it in one step, and confirmed the synthesis of **26** through relevant data, which further confirmed the generation of compound **25**. The HR-MS is shown below: HR-MS (ESI) m/z calcd for $\text{C}_{14}\text{H}_{24}\text{OSeK}$, $[\text{M}+\text{K}]^+$: 355.03932, found: 355.06970.

Q 4: Revising the manuscript, I noticed that the font size on the figures seems a bit small. Please check that the formatting complies with the recommended guidelines.

A 4: Thanks to the reviewer's advice, we have increased the font size on the figures and have changed the number in the figures from 10 to 14.

Q 5: The data availability statement is too vague.

A 5: we greatly appreciated the reviewer's suggestion. For data availability statement, we have changed "The authors declare that the data relating to the characterization of products, experimental protocols and the computational studies are available within the article and its Supplementary Information." to "For full characterization data including NMR spectra of the new compounds and experimental details, see the Supplementary Material. All relevant data underlying the results of this study are

available from the corresponding authors upon request.”

With the above changes and responses, I hope you will find this revised version suitable for publication in *Nature Communications*. If there are any questions, please do not hesitate to inform us. Thank you very much!